# Thermal Analysis of Water-Cooling Permanent Magnet Synchronous Machine for Port Traction Electric Vehicle

**Yongming Tang [1], Shouguang Sun [1], Wenfei Yu [2] and Wei Hua [2],***

1    School of Mechanical, Electronic and Control Engineering, Beijing Jiaotong University, Beijing 100044, China
2    School of Electrical Engineering, Southeast University, Nanjing 210096, China
*    Correspondence: huawei1978@seu.edu.cn

**Abstract:** To further increase the torque/power density of a permanent magnet synchronous machine (PMSM) employed for a port traction electric vehicle, improving the thermal dissipation capacity of the cooling system used in the PMSM has become more and more important. This paper focuses on the thermal analysis of a water-cooling 200 kW PMSM for a port traction electric vehicle. First, the size parameters of the machine and the thermal property parameters of the materials used for each component are given. Based on the heat transfer theory, a fast evaluation method for a transient temperature rise in the water-cooling machine under multiple operating conditions is proposed. A lumped parameter thermal network (LPTN) model is established, and the temperature distributions of the machine at different operating conditions are analyzed. Second, under the same conditions, based on computational fluid dynamics (CFD), a three-dimensional (3D) CFD model is constructed. The influence of different cooling structures on temperature distribution is studied. The validity of the proposed fast evaluation method for a transient temperature rise in water-cooling machines under multiple operating conditions is verified. Finally, the results of the CFD and LPTN calculation are verified by experiments; the maximum temperature deviation of the rated speed/rated power operating condition is 8.5%. This paper provides a reference for the design and analysis of port traction electric vehicle machines.

**Keywords:** permanent magnet synchronous machine; lumped parameter thermal network; computational fluid dynamics; temperature distribution; port traction electric vehicle





## 1. Introduction

In recent years, with the improvement of the torque/power density of permanent magnet synchronous machines (PMSMs) for port traction electric vehicles, the problem of heat dissipation and cooling of the machine has become increasingly prominent [1]. The excessive temperature rise under special working conditions poses a great challenge to the performance and life of the machines [2]. To solve the problem of machine heat dissipation and cooling, it is necessary to start with the cooling structure of the machine and find a suitable heat dissipation structure.

With the ever-increasing demand for torque/power density, research on machine loss and temperature has become a hot topic. An accurate thermal model is an essential tool not only at the machine design stage but also for the prediction of temperature distribution [3]. Recently, ultrasonic vibrations have been introduced as another way to improve heat transfer in thermal systems like water-cooling systems [4,5]. The thermal influence of vehicle integration on the thermal load of a PMSM was discussed by a FEM-based thermal model [6]. In [7], an axially segmented FEM model of an FSPM machine was proposed to analyze the coupled electromagnetic–thermal performances. A numerical approach for the estimation of the convective heat transfer coefficient in the end region of a machine was proposed, and both the local and averaged heat transfer coefficients were estimated [8]. A systematic procedure to study the impact of each thermal phenomenon in interior PM



machines was presented in [9]. In [10], a reduced model in a multi-physical electric machine optimization procedure was proposed.

Considering the temperature variation gradient in armature winding is relatively high, especially for the end-part, a finite element method (FEM) and a thermal circuit coupled model is proposed to investigate the thermal distribution precisely with less time consumption [11,12]. However, the programming procedure of such a coupled model combining FEM and the thermal circuit is complicated and cannot generally be applicable to other machines. Besides, the above-coupled model within thermal analysis only cannot predict the complicated thermal characteristics of electrical machines. While finite element method (FEM)-based and computational fluid dynamics (CFD)-based thermal models can achieve high accuracy, a lumped parameter thermal network (LPTN)-based model has often been preferred thanks to the lower computational effort and good accuracy [13].

The steady-state LPTN method for natural-cooling machines has become increasingly mature. For water-cooling machines, how calculating the convection heat transfer coefficient and accurately calculating the transient temperature rise under multiple operating conditions are still major difficulties. The contribution of this paper lies in the calculation method of the water-cooling convection heat transfer coefficient and the realization of accurate calculation of transient temperature rise under multiple working conditions, which has been verified using computational fluid dynamics and temperature experiments.

In this paper, firstly, taking a 200 kW water-cooling PMSM for port traction electric vehicle as an example, the LPTN model of the machine is built based on heat transfer theory, and the temperature rise in different components of the machine is studied. Secondly, based on the 3D CFD method, the influence of different cooling structures on the machine temperature distribution is studied. Finally, a prototype platform is built, and temperature experiments are carried out to verify the correctness of the theoretical analysis and CFD analysis. This study provides a reference for the design of electric vehicle cooling systems and the study of fluid and temperature fields.

## 2. PMSM Parameters and LPTN Model Building

The research object is a 12 pole 72 slot V-type interior PMSM, and the two-dimensional (2D) cross-section structure and 3D structure of the machine are shown in Figure 1a,b. The peak power and torque of the machine are 200 kW and 2400 Nm. Table 1 shows the main machine parameters. Under a rated operation, the fundamental frequency of the machine current is 150 Hz, and the material of the stator silicon steel sheet is 35WW300. The thermal conductivity of a silicon steel sheet in the x, y, and z directions is 30, 30, and 4.3 W/m·K, respectively, so as to consider the thermal resistance of coating on electrical steel. The machine winding adopts Class H insulation with a maximum heat resistance of 180 °C. The insulation material of the slot is 6650 NHN insulation paper, and the thermal conductivity is 0.2 W/m·K. The type of PMs is N38H, whose maximum energy product is 38MGOe and minimum intrinsic coercivity is 17 kOe. The PM is divided into 15 sections axially to reduce the eddy current loss. The material properties of each machine component are shown in Table 2. The machine is cooled with a housing water jacket, the stator end is not filled, and no additional cooling methods, such as spraying or oil throwing from the rotor shaft hole, are used. The rotor and shaft are also not cooled with oil but mainly rely on the rotor rotating convection heat transfer. Magnetic shear stress is usually used to define the torque density of the machine [14], which value represents the ratio of rated continuous duty torque to rotor volume. The magnetic shear stress value of the PMSM studied in this paper is 66 kNm/m$^3$.

The electromagnetic and loss calculation results of the machine have been carried out in the previous research [15] and will not be repeated here due to the limitation of paper length. The modeling process of the LPTN method considering the water-cooling structure is demonstrated below.

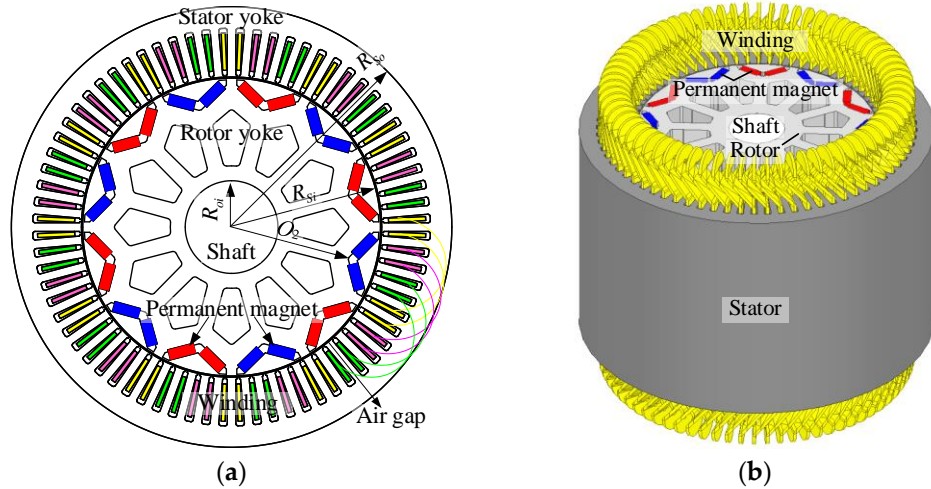

**Figure 1.** The 200 kW PMSM structure. (**a**) 2D Cross Section. (**b**) 3D structure.

**Table 1.** Basic parameters of a 200 kW PMSM.

| Parameters | Values | Units |
|:---:|:---:|:---:|
| Slots | 72 | - |
| Poles | 12 | - |
| Rated power | 130 | kW |
| Rated speed | 1500 | r/min |
| Long-term operation current density | 7.58 | A/mm$^2$ |
| Rated torque | 827.7 | Nm |
| Peak power | 200 | kW |
| Short-term operation current density | 18.7 | A/mm$^2$ |
| Peak Torque @ Speed | 2400 @ 800 | Nm @ r/min |
| Peak Torque duration | 90 | sec |
| Stator outer diameter | 380 | mm |
| Stator inner diameter | 259 | mm |
| Air-gap length | 1 | mm |
| Rotor outer diameter | 257 | mm |
| Rotor inner diameter | 120 | mm |
| Lamination stack length | 240 | mm |

**Table 2.** Material properties of each component.

| Components | Thermal Conductivity/(W/m·K) | | | Specific Heat Capacity/(J/kg·K) | Density/(kg/m$^3$) |
|:---:|:---:|:---:|:---:|:---:|:---:|
| | x | y | z | | |
| Housing | | 168 | | 830 | 2790 |
| Stator Core | 30 | 30 | 4.3 | 460 | 7650 |
| Rotor Core | 30 | 30 | 4.3 | 460 | 7650 |
| PM | | 7.6 | | 460 | 7490 |
| Winding | | 401 | | 385 | 8954 |
| Insulation | | 0.27 | | 1000 | 830 |
| Coolant liquid | | 0.566 | | 4178 | 998 |

Based on the theory of heat transfer, the equivalent thermal network calculates the machine temperature according to the network topology by applying the graph theory to obtain the machine temperature distribution in the way of node temperature.

For the convenience of calculation and improved accuracy, three assumptions are made as follows [16–18]:

(1) Symmetrical temperature distribution and the same cooling conditions along the circumference.

(2) Uniformly distributed thermal capacity and heat generation.

(3) Independent heat flow in radial and axial directions.

Based on the above assumptions, the key thermal nodes of the PMSM machine are demonstrated in the cross-section as shown in Figure 2, where nodes 1–2 represent the machine's internal air, nodes 3–5 represent housing, nodes 6–8 and 14–16 represent stator iron, nodes 17–19 represents rotor, node 9–13 represent armature winding, nodes 20–25 represent rotor yoke, nodes 26–30 represent shaft. Letters a–c represent air outside the housing.

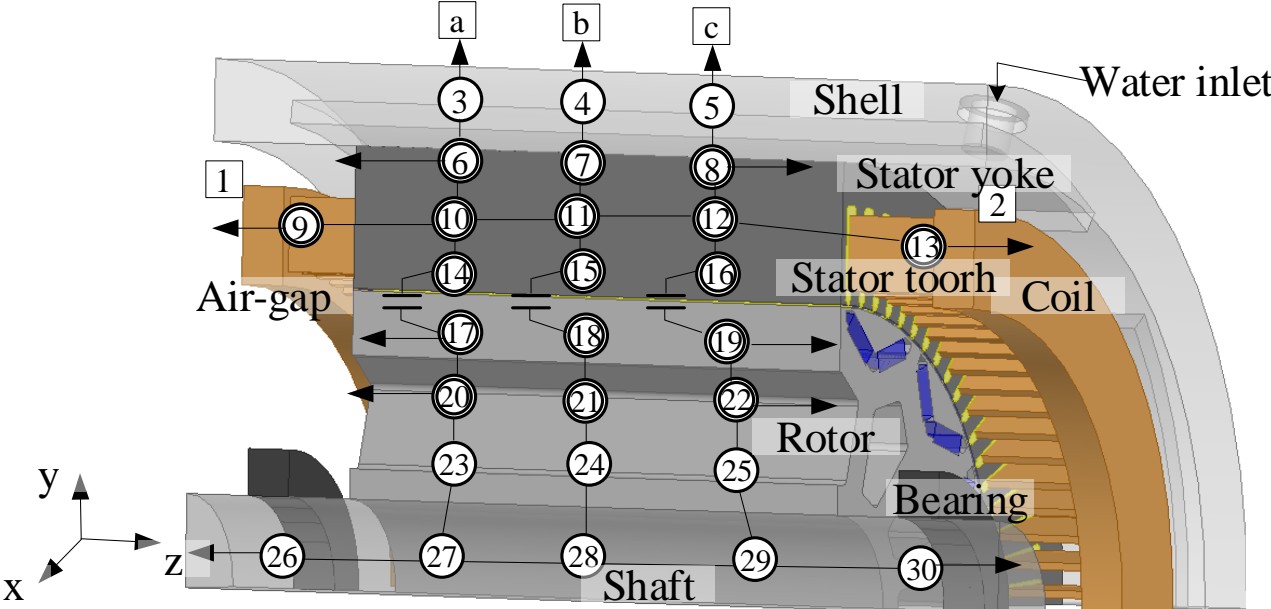

**Figure 2.** The LPTN model of a 200 kW PMSM.

The Reynolds number of the water in the water-cooling channel yields,

$$R_e = \frac{v_{wa}d_{wa}}{v} \tag{1}$$

where $v$ is water kinematic viscosity (m²/s), and $d_{wa}$ is a hydraulic diameter (m). For a rectangular channel, $d_{wa} = 2h_{wa}w_{wa}/(h_{wa} + w_{wa})$, and $h_{wa}$ and $w_{wa}$ are the length and width of the cross-section of the water channel, respectively (m).

If $R_e < 2300$, the water flow state is laminar. For laminar flows, the convective heat transfer coefficient is lower than that of turbulence. The main form of heat transfer is heat conduction. When $R_e > 2300$, the main form of heat transfer is convection.

To enhance the heat transfer from the housing to the water, the flow state is preferably turbulent, where $R_e > 10{,}000$. In this case, the forced water-cooled convection coefficient yields [19],

$$Nu = 0.023R_e^{0.8}P_r^{1/3}\left(\frac{\mu_{wa}}{\mu_{waj}}\right)^{0.14} \tag{2}$$

where $P_r$ is the Prandtl number of water, which changes with self-temperature. For water, $P_r$ decreases with the increase in temperature. $P_r$ decreases from 7 to 1.75 when the water temperature rises from 25 °C to 100 °C. $\mu_{wa}$ and $\mu_{waj}$ are the water dynamic viscosity in the water channel and near the water channel wall (N·s/m²), respectively.

Based on Equations (1) and (2), the convective heat transfer coefficient of the water-cooling structure yields,

$$h_{wa} = Nu \times \frac{\lambda_{wa}}{d} = \frac{0.023\lambda_{wa}p_{Tloss}P_r^{1/3}\mu_{wa}^{0.14}}{2^{0.2}\left(\rho C_{pwa}(T_{outlet} - T_{inlet})v\right)^{0.8}\mu_{waj}^{0.14}} \cdot \frac{(h_{wa} + w_{wa})^{0.2}}{ab} \tag{3}$$

where $\lambda_{wa}$ is the water thermal conductivity (W/(m·K)). $p_{Tloss}$ is the machine's total loss (W). $\rho$ is the water density (kg/m³), $C_{pwa}$ is the water-specific heat capacity (J/(kg·°C)), and $T_{inlet}$ and $T_{outlet}$ are inlet and outlet water temperature, respectively (°C).

The equation for calculating the radial heat conduction resistance of the water-cooling housing is [18]:

$$R_{js} = \frac{R_{Sh} - R_{SO}}{2S_{ss}\lambda_{sh}} \tag{4}$$

The thermal resistance of forced convection heat transfer,

$$R_{fc} = \frac{1}{a_{fc}S_{fc}} \tag{5}$$

The total thermal resistance of node 3-a path is:

$$R_{3a} = R_{js} + R_{fc} \tag{6}$$

where $R_{Sh}$ is the outer radius of the housing (m), $R_{SO}$ is the outer radius of the stator (m), $S_{ss}$ is the effective sectional area of the radial heat conduction of the water-cooling housing (m²), $\lambda_{sh}$ is the thermal conductivity of the water-cooling housing (W/m·K), $a_{fc}$ is the forced convection heat transfer coefficient of the water-cooling housing W/(m²·K), and $S_{fc}$ is the water-cooling area (m²). The equation for calculating the conduction heat resistance between other nodes is similar to Equation (4). The calculated conduction heat resistance of each node is shown in Table 3. The thermal resistance of each contact surface is shown in Table 4.

**Table 3.** Thermal resistance between nodes.

| Nodes | Conduction Thermal Resistance (K/W) | Nodes | Conduction Thermal Resistance (K/W) |
|---|---|---|---|
| 3–a, 4–b, 5–c | $6.59 \times 10^{-4}$ | 14–15, 15–16 | 0.127 |
| 3–4, 4–5 | 4.14 | 14–17, 15–18, 16–19 | $1.93 \times 10^{-4}$ |
| 3–6, 4–7, 5–8 | $3.5 \times 10^{-3}$ | 17–18, 18–19 | 0.1429 |
| 6–7, 7–8 | 0.1133 | 17–20, 18–21, 19–22 | 0.098 |
| 6–10, 7–11, 8–12 | 0.1197 | 20–21, 21–22 | 0.0128 |
| 9–10, 12–13 | 0.1197 | 20–23, 21–24, 22–25 | 0.0082 |
| 1–9, 2–13 | 0.0672 | 23–24, 24–25 | 0.0108 |
| 10–11, 11–12 | 0.0149 | 23–27, 24–28, 25–29 | 0.0092 |
| 10–14, 11–15, 12–16 | 0.0156 | 26–27, 27–28, 28–29, 29–30 | 0.0083 |

**Table 4.** Thermal resistances of interfaces between umpteen components.

| Items | Gap (mm) | Thermal Conductivity (W/m·K) | Thermal Resistances (K/W) |
|---|---|---|---|
| magnets/rotor | 0.1 | 0.03171 | 0.003153 |
| stator/housing | 0.015 | 0.03171 | 0.000473 |
| housing/coolant tubes | 0.005 | 0.03171 | 0.0001577 |

The nodal steady-state temperature distribution $T$ is calculated with the column vector of nodal power loss $W$ and the inter-nodal conductance matrix $G$ [20–22]. Matrix $G$ is formed using the thermal conductivities of every path as,

$$
G = \begin{bmatrix}
\sum_{k=1}^{N} \frac{1}{R(k,1)} & -\frac{1}{R(2,1)} & \cdots & -\frac{1}{R(N,1)} \\
-\frac{1}{R(2,1)} & \sum_{k=1}^{N} \frac{1}{R(k,2)} & \cdots & -\frac{1}{R(N,2)} \\
\vdots & \vdots & \vdots & \vdots \\
-\frac{1}{R(N,1)} & -\frac{1}{R(N,2)} & \cdots & \sum_{k=1}^{N} \frac{1}{R(k,N)}
\end{bmatrix}
\tag{7}
$$

where $R$ means the thermal resistance between two nodes, and $N$ means the order of $G$ (it is 30 here).

Thus, the nodal steady-state temperature distribution and transient temperature rises can be expressed, respectively, as:

$$
T = G^{-1}W
\tag{8}
$$

$$
\frac{dT}{dt} = (MC)^{-1}W - (MC)^{-1}GT
\tag{9}
$$

where matrix $M$ is the mass of materials (kg), matrix $C$ is the specific heat (J/(kg·K)), and both matrices are $30 \times 30$ order. The column vector $T$ is the temperature distribution of each node (K).

In general, only the vector $T$ changes with time $t$. To solve Equation (9), time $t$ is set as a small step, so the continuity equation is discretized, and the status of $T(k + 1)$ can be realized by $T(k)$. The discretization may result in non-convergence and time consumption given an improper small step. Therefore, the analytical solution of the matrix differential equation is derived as,

$$
T(t) = eA^{(t-t0)}T(t_0) + A^{-1}\left(e^{A(t-t0)} - I\right)B
\tag{10}
$$

where $I$ is a unit matrix, $t_0$ is set as 0 in the following analysis, and matrix $A$ and vector $B$ can be obtained by,

$$
\begin{cases}
A = -(MC)^{-1}G \\
B = (MC)^{-1}W
\end{cases}
\tag{11}
$$

Hence, Equation (8) gives the solution to steady-state temperature distribution, and Equation (10) supplies the solution to transient temperature rise. The above solutions can be realized by programming based on commercial software, such as Matlab.

Based on the LPTN model, the transient temperature corresponding to the three typical operating conditions of the machine is shown in Figure 3a–c. The three typical operating points are rated speed/rated power operating point (1500 r/min @ 130 kW), peak torque/peak power operating point (796 r/min @ 200 kW), and peak speed/rated power operating point (2700 r/min @ 130 kW), respectively. The operating time corresponding to short-duration torque is 90 s. It can be seen from Figure 3 that due to the small thermal resistance from the stator side components to the housing, the heat transfer efficiency is high, and the stator components enter the thermal balance state after continuous operation for 20 min. However, the heat dissipation effect on the rotor side is poor, and it takes about 120 min to reach the thermal balance, which applies to different working conditions.

The temperature of the housing, stator, rotor, PM, coolant, and winding end is 62, 93.4, 86.8, 85.7, 58.2, and 110.5 °C, respectively, after continuous operation at a rated speed/rated power for 120 min. The copper loss is the main loss under the peak torque/peak power operation condition, and the winding insulation class restricts the continuous operation time of the machine. When the continuous operation time reaches 120 s, the winding temperature reaches 175.8 °C. Due to the machine adopting class H insulation, the temperature

does not exceed the insulation withstand temperature, which does not affect the normal operation of the machine. As the operating frequency and field weakening current increase at the peak speed/rated power operating condition, the stator iron loss and copper loss also increase. The temperature of the housing, stator, rotor, PM, coolant, and winding end is 68.7, 131.7, 120.6, 117.6, 61.3, and 158.1 °C, respectively.

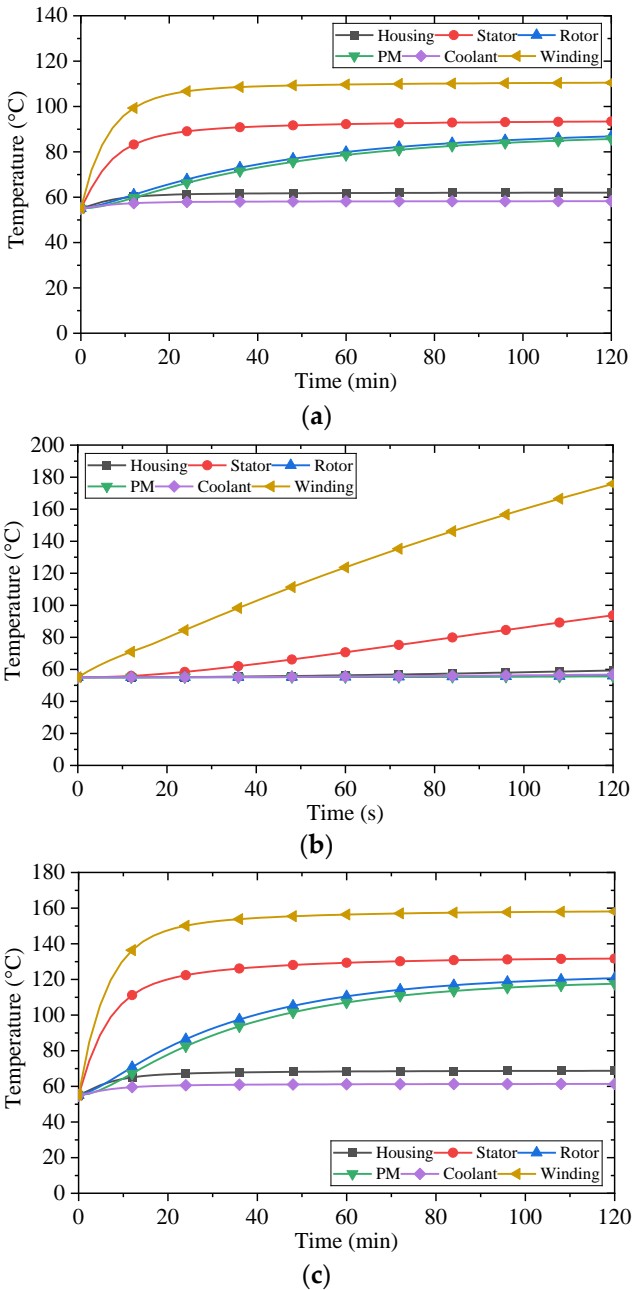

**Figure 3.** The transient temperature under three typical operating conditions. (**a**) 1500 r/min @ 130 kW. (**b**) 796 r/min @ 200 kW. (**c**) 2700 r/min @ 130 kW.

## 3. CFD Method

For the machine water-cooling structure, the water injection forms mainly include a single input and a single output, single input and double output, double input, and double output, etc. The structure mainly includes axial spiral-type, axial S-type, axial Z-type, radial H-type, etc. Three kinds of water channel structures are designed for the 200 kW PMSM, as shown in Figure 4a–c, namely axial spring-type, axial S-type, circular S-type,

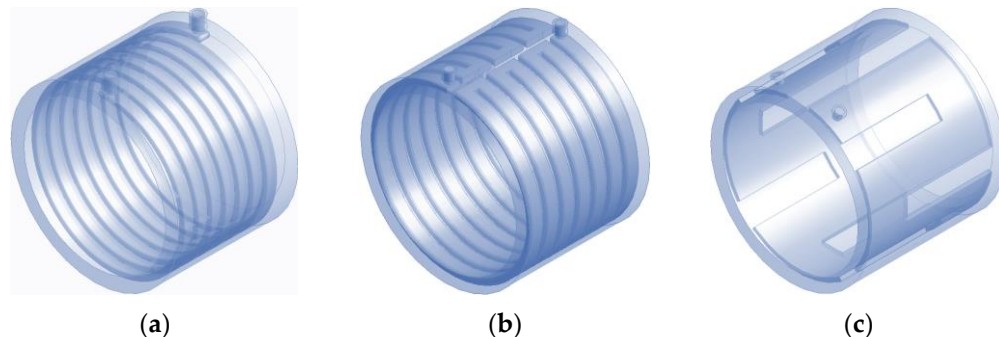

**Figure 4.** Three kinds of water channel structures. (**a**) Axial spiral type. (**b**) Axial S type. (**c**) Circumferential S type.

Based on the CFD method and commercial software Ansys platform, geometric simplification, Mesh generation, and other pre-processing are carried out for the 3D machine thermal model. The 1:1 temperature field CFD model and grid generation are established, as shown in Figure 5. The number of nodes is 17,229,841, and the number of cells is 7,836,166. The average quality of the grid is 0.87. The aspect ratio is 2.03. Through this part of the research, on the one hand, the cooling effects of different structures can be obtained, and on the other hand, the temperature calculation results of the previous LPTN model can be verified.

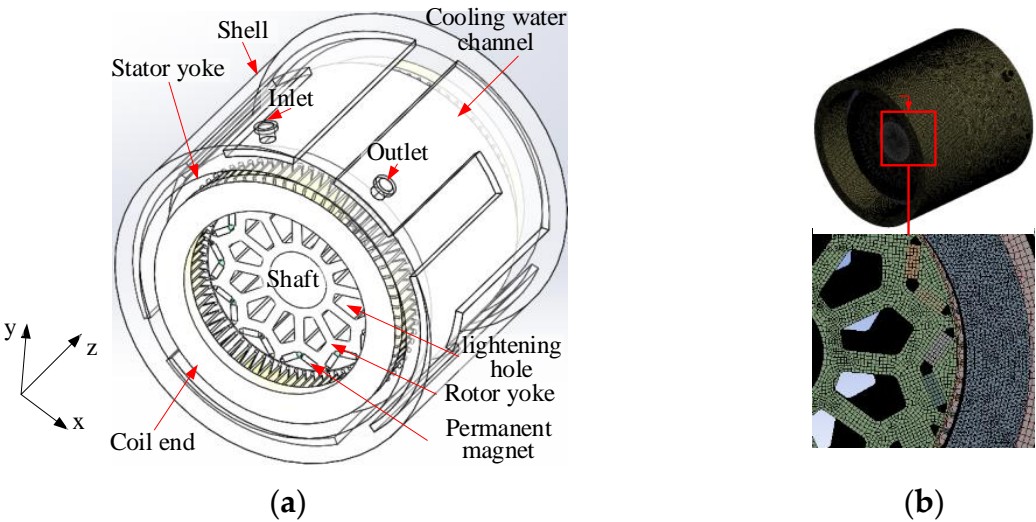

**Figure 5.** CFD model and mesh generation. (**a**) CFD model. (**b**) Mesh.

In order to compare and study the heat dissipation capacity of three cooling structures, the same main heat source is adopted, including a stator, rotor, winding, and PM heat source. Fluid water is used as the coolant, and the ambient pressure is the standard atmospheric pressure. The inlet coolant temperature of the three models is 55 °C, and the inlet coolant flow is 20 L/min. Figure 6a–c show the steady-state temperature distribution of three water channel structures' under the rated speed/rated power, respectively.

The highest temperature location of the machine is at the end of the winding. By comparing the three cooling structures, it is found that the third cooling structure has the best cooling effect. The reason is that the axial length of the machine is large, and the length-diameter ratio is close to one. The temperature difference between the water inlet and outlet of the axial spiral-type and axial S-type structure is large, resulting in a large difference in the axial temperature inside the machine, and the cooling effect of the end temperature of one side of the winding is slightly poor. The coolant temperature of the circumferential S-type structure flows in the axial direction, and the temperature distribution is uniform,

which can exchange the heat inside the machine in time. Therefore, the prototype adopts the third cooling structure. Based on the circumferential S-type structure, the transient temperature changes under three typical working conditions calculated using the CFD method are shown in Figure 7a–c.

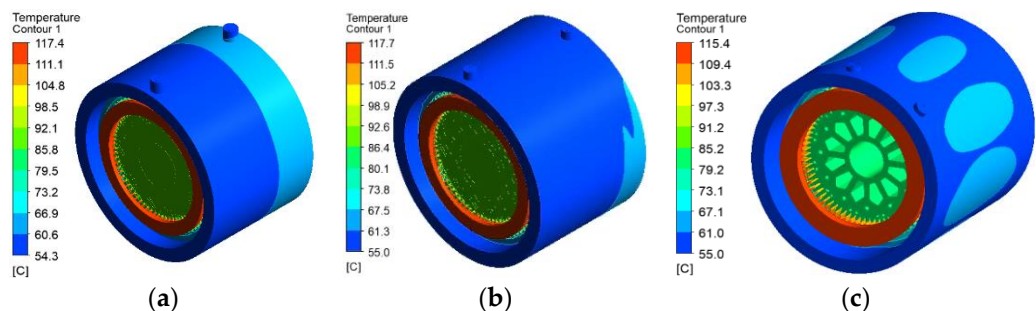

**Figure 6.** Machine temperature distributions. (**a**) Axial spiral type. (**b**) Axial S type. (**c**) Circumferential S type.

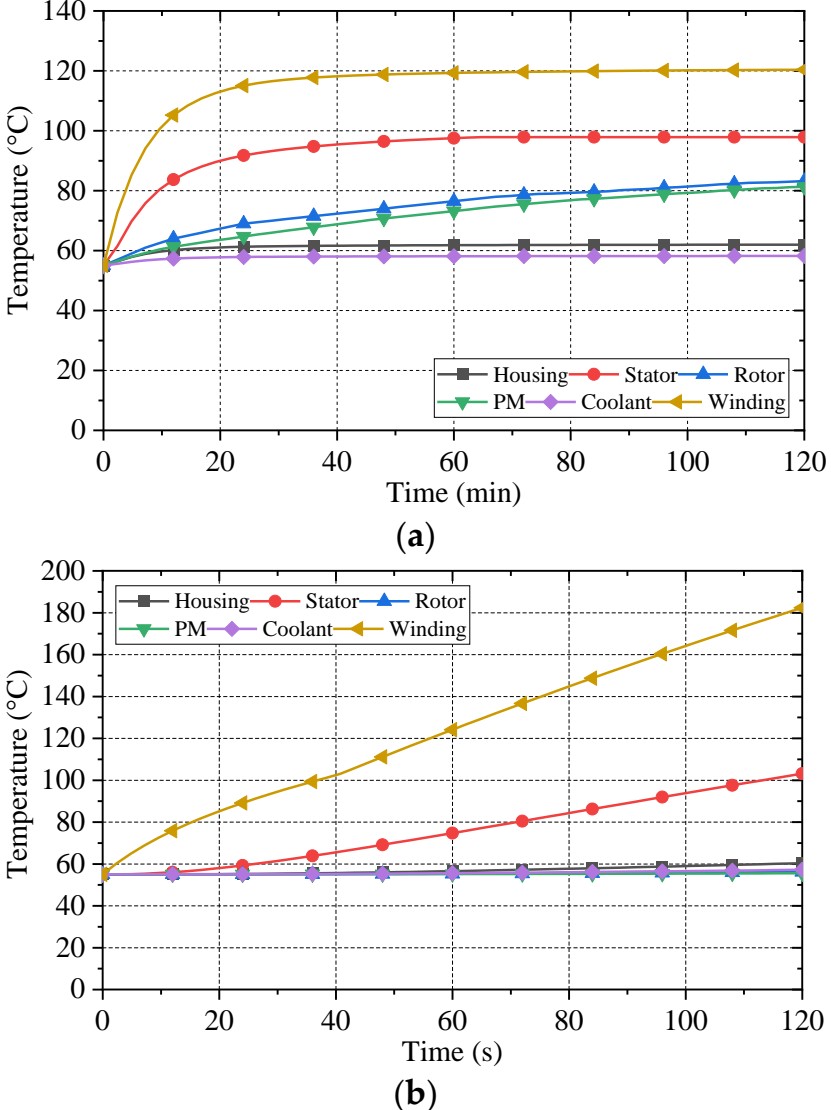

**Figure 7.** *Cont.*

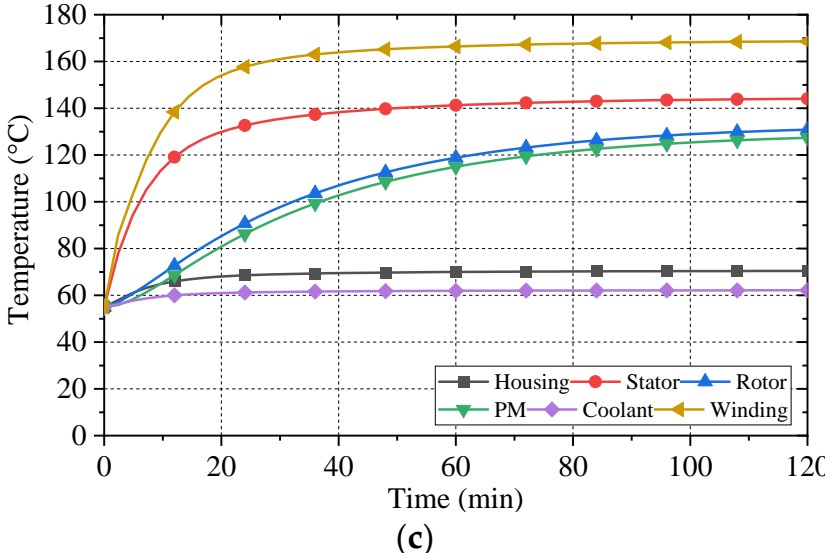

**Figure 7.** The transient temperature under three typical operating conditions calculated using the CFD method. (**a**) 1500 r/min @ 130 kW. (**b**) 796 r/min @ 200 kW. (**c**) 2700 r/min @ 130 kW.

In Figure 7, the machine temperature rises rapidly due to the large heat source inside the machine during the load operation, and the temperature of stator components reaches stability quickly. It can be seen from the transient temperature rise process in Figure 7a,c that the temperature of stator components is stable after 20 min. The temperature of the rotor reaches stability in 2 h. There are two main reasons: on the one hand, the loss of the rotor is small, and the speed of heat generation and heat dissipation of the rotor to reach the thermal balance is slow. On the other hand, due to the poor heat conduction of the air gap, the heat exchange rate between the stator and the rotor is low, which can be seen from the LPTN analysis in Section 2.

The copper loss is the main loss under the peak torque/peak power operation condition, and the winding temperature rises sharply. After continuous operation for 120 s, the winding temperature reaches 182.3 °C. As there is only slot insulation and winding insulation paint between the stator core and the winding, the temperature of the stator rises to 103.1 °C after 120 s of continuous operation, and the temperature rise of other rotor parts is not obvious due to the large thermal resistance between winding heat sources.

Tables 5–7 compare the temperature calculation results of the LPTN method and the CFD method under three typical working conditions. It can be seen that the maximum error of the two methods is within 9%, which can verify the correctness of the calculation methods.

**Table 5.** Comparison of temperature calculation results under 1500 r/min @ 130 kW.

| Components | LPTN Results (°C) | CFD Results (°C) | Deviation (%) |
|:---:|:---:|:---:|:---:|
| Housing | 62.0 | 62.1 | 0.16 |
| Stator | 93.4 | 97.9 | 4.6 |
| Rotor | 86.8 | 83.2 | 4.33 |
| PM | 85.7 | 81.3 | 5.41 |
| Coolant | 58.2 | 58.3 | 0.17 |
| Winding | 110.5 | 120.4 | 8.22 |

**Table 6.** Comparison of temperature calculation results under 796 r/min @ 200 kW.

| Components | LPTN Results (°C) | CFD Results (°C) | Deviation (%) |
|---|---|---|---|
| Housing | 59.2 | 60.3 | 1.82 |
| Stator | 93.7 | 103.1 | 9.12 |
| Rotor | 55.9 | 56.1 | 0.36 |
| PM | 55.5 | 55.6 | 0.18 |
| Coolant | 56.7 | 57.2 | 0.87 |
| Winding | 175.8 | 182.3 | 3.57 |

**Table 7.** Comparison of temperature calculation results under 2700 r/min @ 130 kW.

| Components | LPTN Results (°C) | CFD Results (°C) | Deviation (%) |
|---|---|---|---|
| Housing | 68.7 | 70.4 | 2.41 |
| Stator | 131.7 | 144.1 | 8.61 |
| Rotor | 120.6 | 130.9 | 7.87 |
| PM | 117.6 | 127.4 | 7.69 |
| Coolant | 61.3 | 62.1 | 1.29 |
| Winding | 158.1 | 168.6 | 6.23 |

## 4. Experimental Study

To verify the correctness of the above two temperature evaluation methods, a 200 kW PMSM prototype is processed, and a prototype test platform is built. The stator, rotor, rotor components, and prototype assembly photos are shown in Figure 8, and the prototype temperature rise test platform is shown in Figure 9. The experimental equipment includes a dynamometer, power analyzer, high-precision bidirectional DC power supply, DSP controller, industrial water-cooling device, temperature patrol instrument, etc. The DC power supply adopts Kewell's EVD-300-800, with a rated output power of 300 kW, output DC voltage range of 24~800 V, and output current of −700~700 A. The power analyzer is YOKOWAGA WT3000E, the measurement bandwidth is DC and 0.1 Hz~1 MHz, and the accuracy is ±0.02% of the reading. Thermistor PT100 is placed at the end coil and stator core to monitor the temperature of the winding end and stator core during machine operation. During the load temperature rise test, the ambient temperature is 19 °C, the coolant flow rate is 20 L/min, and the initial coolant temperature is 55 °C.

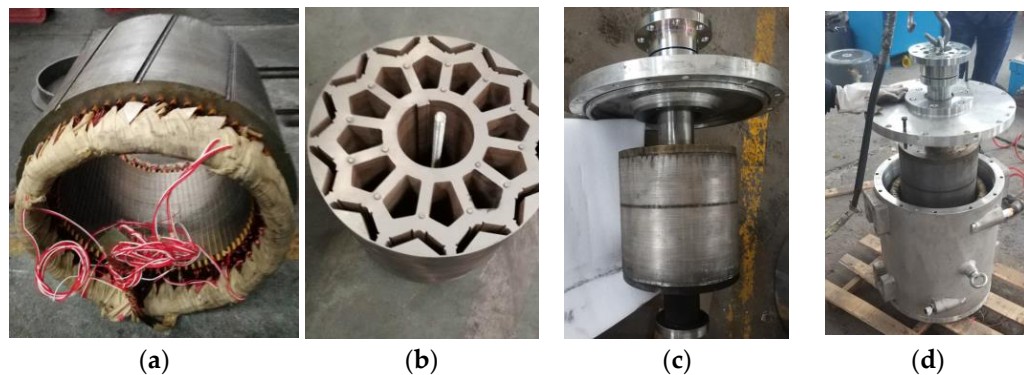

      (**a**)                  (**b**)                  (**c**)                  (**d**)

**Figure 8.** Machine components. (**a**) Stator assembly. (**b**) Rotor core. (**c**) Rotor assembly. (**d**) Machine assembly.

The machine temperature is measured based on the temperature rise test platform in Figure 9 when the initial conditions for temperature calculation are consistent. In the experiment, the temperature rises of rated speed/rated power working point and peak torque/peak power point are tested. From the previous analysis, it can be seen that the winding and stator can reach thermal stability after continuous operation for about 20 min. Therefore, the temperature rise test for the rated load is recorded continuously for 22.5 min

in the experiment. The test results are shown in Figure 10a. The maximum temperature deviation of the rated speed/rated power operating condition is 8.5%. By comparing the transient temperature of different temperature calculation methods, it can be seen that the dynamic cumulative deviation does not exceed 10%, and the correctness of the temperature calculation method is verified using experiments.

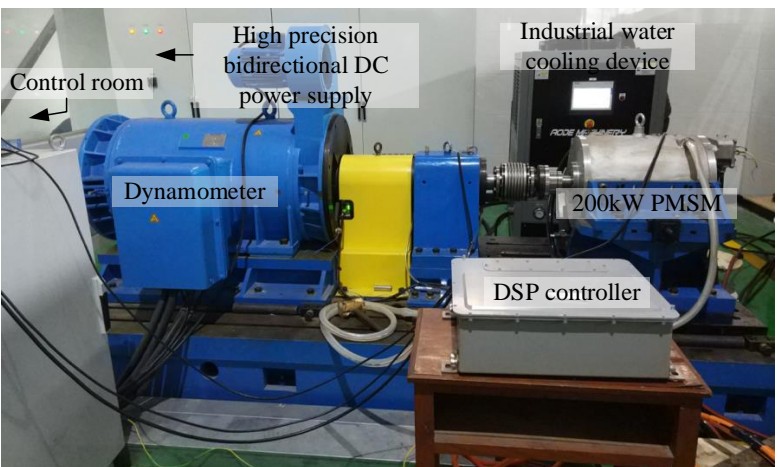

**Figure 9.** Prototype temperature rise test platform.

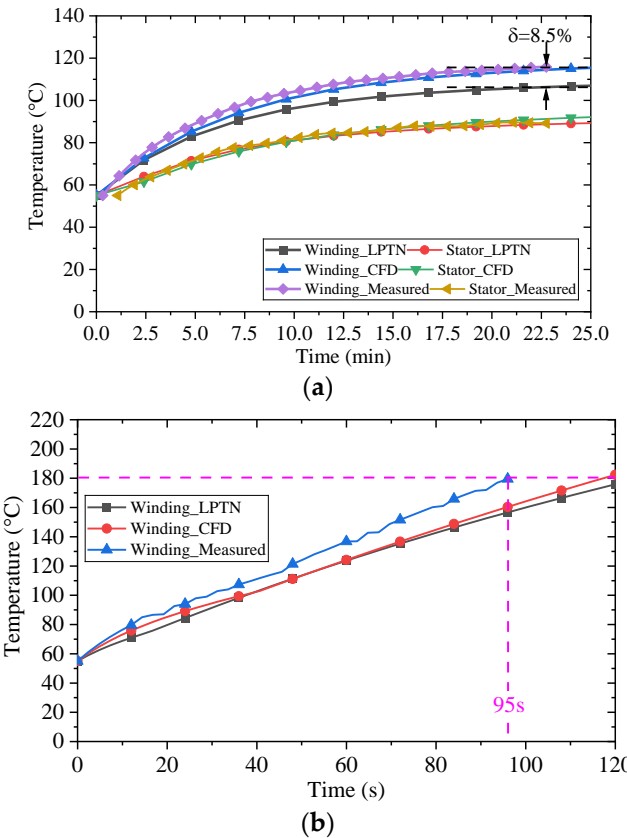

**Figure 10.** Comparison of experimental temperature results with calculated results. (**a**) 1500 r/min @ 130 kW. (**b**) 796 r/min @ 200 kW.

The temperature test results and calculation results at the peak torque/peak power operating condition are shown in Figure 10b. It can be seen that the result of the two temperature calculation methods is that the winding temperature can reach 180 °C only after 120 s of

continuous operation. When the prototype runs for 95 s, the winding end temperature reaches 180 °C. At this time, the power supply is cut off to avoid a winding fault. The prototype winding adopts a Y-type connection mode, and the neutral point is welded and fixed before being buried in the winding end wire package. Due to the manual welding method, the welding quality effect is not very good, which causes a change in local resistance. There is no oil cooling or potting, and there is uneven temperature distribution at the winding end. These two reasons lead to a large deviation between the measured winding temperature and the calculation.

### 5. Conclusions

This paper proposes a temperature evaluation method of a 200 kW water-cooling PMSM for a port traction electric vehicle. Based on the LPTN and CFD methods, the transient temperature calculation under multiple working conditions is studied and experimentally verified using the temperature test platform. The conclusions can be drawn as follows:

(1) For the water-cooling PMSM studied in this paper, the temperature of stator components is stable after 20 min. The reason is that the thermal resistance from the stator side parts to the shell is small, and the heat transfer efficiency is high. The temperature stability of the rotor part takes 2 h. There are two reasons: a. Due to the small rotor loss, the heating and heat dissipation speed of the rotor is very slow, and it takes a long time to reach the thermal balance. b. Due to the poor heat conduction effect of the air gap, the heat exchange rate between the stator and rotor is poor.

(2) The highest temperature location of the machine is at the end of the winding. By comparing the three cooling structures, it is found that the third cooling structure has the best cooling effect. The reason is that the axial length of the machine is large, and the length-diameter ratio is close to one. The temperature difference between the water inlet and outlet of the axial spiral type and axial S-type structure is large, resulting in a large difference in the axial temperature inside the machine, and the cooling effect of the end temperature of one side of the winding is slightly poor. The coolant temperature of the circumferential S-type structure flows in the axial direction, and the temperature distribution is uniform, which can exchange the heat inside the machine in time.

(3) The maximum temperature deviation of the rated speed/rated power operating condition is 8.5%. By comparing the transient temperature of LPTN, CFD, and experimental methods, it can be seen that the dynamic cumulative deviation does not exceed 10%.

**Author Contributions:** Conceptualization, W.H.; Methodology, Y.T.; Software, S.S.; Investigation, W.Y. All authors have read and agreed to the published version of the manuscript.

**Funding:** This research was funded by the Key R&D Program of Jiangsu Province grant number BE2022032.

**Data Availability Statement:** The data presented in this study are available on request from the corresponding author. The data are not publicly available due to the project is still in progress.

**Conflicts of Interest:** The authors declare no conflict of interest.

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
