# Peer review of "Thermal Analysis of Water-Cooling Permanent Magnet Synchronous Machine for Port Traction Electric Vehicle"

_electronics, doi:10.3390/electronics12030734_

Round 1
Reviewer 1 Report
Thermal management (including heat sources, cooling channel structures, and thermal resistances) is known to be central in the design of torque-dense motors. In this context, the author's paper could have been a welcome contribution to the available body of literature on the subject. But unfortunately, this 200 kW motor design does not qualify to be called a ‘torque-dense motor’, unless significant improvements in its electrical and thermal design of the motor are made, as per my following observations.
line 29 – torque density.
Torque-dense motors are judged by their magnetic shear stress s (magnetic shear force/rotor surface area) at the air gap
where T = rated continuous duty torque, D = outer diameter of the rotor, and L = stack length. Per the textbook by Hendershot and Miller, Design of brushless permanent magnet motors, the maximum s achieved so far by liquid-cooled motors is 15 psi (100 kNm/m3). [I know that private industry has achieved higher shear stress of 40 psi, but let me limit to info available in the public domain]. For the authors’ motor, T = 827.7 Nm, D = 257mm, L = 240mm, so s = 4.8 psi (33kNm/m3). This is so low compared to 15 psi. So I do not consider this a torque-dense motor. Significant changes in the electrical and thermal design have to be made to achieve a ‘torque-dense motor’ status.
Designing a torque-dense motor is a complex struggle between efficiency vs current density and gap flux density. Generally designer clamps the efficiency during the preliminary stages of design. This fixes a target heat load. Then you make iterations on electrical and thermal designs to meet the target efficiency requirement. In electrical design, you pick steel, magnet wire, and magnets You try to increase the current density and gap flux density to increase the torque density, but increasing them too much produces a lot more heat which reduces efficiency. In thermal design, you identify hot spots and design the ducting to pull heat from them with minimal thermal resistance into the air.
Unfortunately, I did not find key parameters of electrical design, viz. efficiency, conductor current density, slot current density, gap flux density, and operating points (f, B) in laminations and magnets both in continuous duty and short duty. The author did document short-duration and continuous duty temperature rise as well as an overview of coolant channel structure types. But I did not find key parameters of thermal design, viz. hot spots, allowable temperature rise, heat transfer paths, cooling channels in end windings and rotors, materials used in slot liners, the thermal resistance of coatings on electrical steels, thermal resistances of interfaces between umpteen components such as magnets/rotor, stator/housing, housing/coolant tubes, etc. I would advise that the author should provide information on some of these details to allow one to make a judgment about whether it is a torque-dense motor or not.
line 75 – peak torque 2400 Nm
The authors are referring to what is known as short-duration torque. If so they should specify the duration say 20 sec. Otherwise, it is incomplete.
line 77 - 35WW300 steel
The choice of steel is critical in designing high-torque-density motors. I guess the authors’ selection of 35WW300 is adequate if cost is a driving factor, as it is better than other 0.35mm class steels. (its core loss is 15w/kg at 1.5T, 200 Hz operating point, better than the costlier B35A230 which produces 16.9 w/kg at the same operating point). The authors could reduce the core loss if they use thinner gauge steels such as NO20 or NO30. What prevented you from using them to reduce the heat load?
Table 1
peek torque – replace it with peak torque
‘Silicon steel sheet thickness’ – I think the authors refer to ‘lamination stack length’.
Table 2
‘cooling liquid 0.566 w/mK’ – liquids are not characterized by k by heat transfer coefficient. I am also curious why authors are using water as a coolant. It freezes at 0oC. Folks normally use antifreeze or oil as the coolant in motors to operate over a wider operating temperature range.
Fig. 2
It does not show the x,y, or z axis, not sure whether the z-axis corresponds to the axis of rotation of the motor. It does not show where the coolant enters, where it exits, or how it flows through to pull out the heat. In torque-dense motors, it is very critical to remove heat from end windings in the stator and the hot magnets in the rotor. I do not think I found any information on them in this paper. The author should include a couple of paragraphs on how they pull out heat from the end windings and the rotor.
line 181 PM temperature
authors say that PM reached 117.6C under short duration (‘peak operation condition’). But it is well known that temperature distribution in magnets is highly non-uniform. Generally, the ends of magnets will be at a higher temperature, depending upon the design. The authors should provide information on the temperature distribution in magnets, as otherwise, they can fail.
line 330 – Conclusion ‘The copper wires at the end of the prototype are not filled with insulating paint, which restricts the operation time of the over-load operation condition’. This ‘conclusion’ is not something that is not known, as motor designers have recognized in the past that cooling of end windings is critical in the design of motors (for example the well-known Prius Hybrid electric drive motor has gone through several design changes between 2000 to 2010 to address cooling of end windings). Similarly, authors will learn that, unless the rotor magnets are liquid-cooled, their hot spot temperatures will be high, forcing motors to operate at lower torque, thereby lowering the torque density.
Author Response
To Reviewer:
Manuscript ID: electronics-2159933. Thermal Analysis of Water-Cooling Permanent Magnet Synchronous Machine for Port Traction Electrical Vehicle
My co-authors and I would like to thank the reviewers and the editor for their constructive and helpful comments and suggestions. All of their comments and suggestions have been taken into full consideration in this revised version.
All changes are highlighted in RED in this revised version. The detailed responses to reviewers’ comments are enclosed.
Yours Sincerely,
Dr. Wei Hua
School of Electrical Engineering
Southeast University
Nanjing 210096
China
Email: huawei1978@seu.edu.cn

Reviewer 2 Report
Manuscript ID: electronics-2159933
Title: Thermal Analysis of Water-Cooling Permanent Magnet Synchronous Machine for Port Traction Electrical Vehicle
In this study, the thermal analysis of a water-cooling 200kW PMSM for a port traction electrical vehicle is investigated. The quality of the figures is perfect. The results are verified successfully. The following minor and major comments should be addressed:
· Check the language and revise typos and grammatical errors.
· Enrich the abstract and conclusion sections with more quantitative data from the research.
· In eq. 2: Re is Reynolds number or Rayleigh number?
· The acceptable range of Reynolds and Prandtl numbers should be stated for using equation 2. This limitation is stated in Incorp’s Heat Transfer book.
· In the line after eq. 2: “ μwa and μwaj is the water dynamic viscosity, respectively.” This statement must be edited.
· Incropera's Principles of Heat and Mass Transfer, 8th Edition, Global Edition
Frank P. Incropera, David P. DeWitt, Theodore L. Bergman, Adrienne S. Lavine
· Recently, ultrasonic vibrations have been introduced as another way to improve heat transfer in thermal systems like water-cooling systems. It is suggested to mention this in the introduction for readers’ information. Referring to the following articles will be helpful: Doi: 10.3390/fire6010013; Doi: 10.3390/w14244000.
· The mesh grid study should be done for the numerical simulations.
· It is necessary to explain the characteristics of the test equipment in more detail.
Author Response

(The authors gave the same response as above.)

Reviewer 3 Report
This paper deals with the thermal analysis of water-cooling permanent magnet synchronous machine for port traction electrical vehicle based on a lumped parameter thermal network and on 3-D computational fluid dynamics. The subject is very interesting, the introduction is satisfactory, the results are promising, but some points of the mathematical base should be explained in a better way. More specifically:
è All parameters should be described, such as “μwa”, “μwa”, “d”, “pTloss”, “a”, “b”, “ρ”, “Cpwa”, “Sss”, as well as their units etc. Similarly the formation of the matrices W and C such as G one.
è In CFD method the CFD tool should be described, as well as the population of mesh etc.
è In results we can see differences between simulation results and experiment. The superiority of CFD from lumped parameter thermal network is obvious in Fig. 10(a), but in Fig. 10(b) the difference between experimental results and CFD is big. Why does this happen?
Author Response

(The authors gave the same response as above.)

Round 2
Reviewer 2 Report
All the comments are adressed.
Author Response
To Reviewer:
Thank you very much.
Yours Sincerely,
Dr. Wei Hua
School of Electrical Engineering
Southeast University
Nanjing 210096
China
Email: huawei1978@seu.edu.cn
Reviewer 3 Report
The authors have answered in all remarks, except the first one. "All parameters should be described, such as “μwa”, “μwa”, “d”, “pTloss”, “a”, “b”, “ρ”, “Cpwa”, “Sss”, as well as their units etc. Similarly the formation of the matrices W and C such as G one." Because i.e. eq. (4) parameter "Sss" is presented while in line 145 "Sjs" is presented. Other parameters, i.e. "pTloss” have not been explained. The authors should examine every variable of equations , tables etc. and they should refer / analyze/ describe the respective variable.
Author Response
To Reviewer:
Manuscript ID: electronics-2159933. Thermal Analysis of Water-Cooling Permanent Magnet Synchronous Machine for Port Traction Electrical Vehicle
My co-authors and I would like to thank the reviewer for your constructive and helpful comments and suggestions. All of the comments and suggestions have been taken into full consideration in this revised R2 version. All the variable of equations, tables etc. have been explained in the manuscript.
All changes are highlighted in RED in this revised version. The detailed responses to reviewers’ comments are enclosed.
Yours Sincerely,
Dr. Wei Hua
School of Electrical Engineering
Southeast University
Nanjing 210096
China
Email: huawei1978@seu.edu.cn
